# How long is enough to detect terrestrial animals? Estimating the minimum trapping effort on camera traps

Xingfeng Si[1], Roland Kays[2] and Ping Ding[1]

[1] The Key Laboratory of Conservation Biology for Endangered Wildlife of the Ministry of Education, College of Life Sciences, Zhejiang University, Hangzhou, Zhejiang, China
[2] North Carolina Museum of Natural Sciences & NC State University, Raleigh, NC, USA

## ABSTRACT

Camera traps is an important wildlife inventory tool for estimating species diversity at a site. Knowing what minimum trapping effort is needed to detect target species is also important to designing efficient studies, considering both the number of camera locations, and survey length. Here, we take advantage of a two-year camera trapping dataset from a small (24-ha) study plot in Gutianshan National Nature Reserve, eastern China to estimate the minimum trapping effort actually needed to sample the wildlife community. We also evaluated the relative value of adding new camera sites or running cameras for a longer period at one site. The full dataset includes 1727 independent photographs captured during 13,824 camera days, documenting 10 resident terrestrial species of birds and mammals. Our rarefaction analysis shows that a minimum of 931 camera days would be needed to detect the resident species sufficiently in the plot, and *c.* 8700 camera days to detect all 10 resident species. In terms of detecting a diversity of species, the optimal sampling period for one camera site was *c.* 40, or long enough to record about 20 independent photographs. Our analysis of evaluating the increasing number of additional camera sites shows that rotating cameras to new sites would be more efficient for measuring species richness than leaving cameras at fewer sites for a longer period.

Corresponding author
Ping Ding, dingping@zju.edu.cn

## INTRODUCTION

Camera traps have become an important tool for inventorying elusive wildlife across a variety of habitats (*Cutler & Swann, 1999*; *Silveira, Jacomo & Diniz-Filho, 2003*; *O'Connell, Nichols & Karanth, 2011*). These data can be used for animal community metrics (*Tobler et al., 2008*), population abundance indices (*O'Brien, Kinnaird & Wibisono, 2003*), and sometimes, absolute density (*Karanth & Nichols, 1998*; *Silver et al., 2004*). Despite their popularity, surprisingly few studies have examined survey results to offer guidelines for efficient study design (*Rowcliffe et al., 2008*; *Tobler et al., 2008*; *Rovero & Marshall, 2009*; but see *Williams, Nichols & Conroy, 2002*; *MacKenzie et al., 2006*). The consequence is that

new surveys are designed without guidance or following untested rules of thumb about the sampling effort needed to reach their goals.

One key metric that is useful for planning efficient studies documenting diversity of wildlife in a particular area is the minimum trapping effort (MTE). MTE is the trapping effort–the number of camera days–required to record the species of interest in a specific area (*Yasuda, 2004*). Estimates of MTE allow the design of an efficient inventorying plan that does not extend too long, nor fail to detect some species present in the survey area. The trapping efforts reported in the literature varies widely (*Maffei, Cuéllar & Noss, 2004*; *Wegge, Pokheral & Jnawali, 2004*; *Trolle & Kéry, 2005*; *Li et al., 2010*). Even for the same species, the MTE varies widely across studies, for example, the recommended trapping efforts varied from 450 camera days in Bolivia (*Trolle & Kéry, 2003*) to 2280 in Brazil (*Maffei et al., 2005*) for targeting the ocelot (*Leopardus pardalis*). Factors that might affect the MTE include habitat, weather conditions, abundance of target species, and sampling strategies such as the use of baits and spacing of cameras (*Wegge, Pokheral & Jnawali, 2004*; *Rovero & Marshall, 2009*). Furthermore, a variety of different statistics have been used to estimate MTE, including species accumulation curves (*Wegge, Pokheral & Jnawali, 2004*; *Azlan, 2006*), calculations based on the probability of capture (*Tobler et al., 2008*), simulation models (*Rowcliffe et al., 2008*), sampling precision analysis (*Rovero & Marshall, 2009*) and bootstrap procedures based on the precision of parameter estimation (*Goswami et al., 2012*). Finally, there is little distinction between the number of camera days and the number of camera sites, in these studies.

Similar to the MTE in studies of camera traps, the minimal area of a plant community is a classic concept introduced in the 1920s (*Hopkins, 1957*), and regularly discussed since then (*Goodall, 1952*; *Hopkins, 1957*; *Whittaker, 1980*; *Barkman, 1989*). The related curve, known as the species accumulation curve, is the relationship of the number of species and the sampling effort, which may depend on the time or area sampled. One expects curves to approach an asymptote, and thus give a judgment of sampling adequacy (*Daubenmire, 1968*). In long-term monitoring projects, sampling over gradients in time is logically similar to sampling over gradients in space (*Colwell & Coddington, 1994*). Thus, for camera traps, the relationship between trapping efforts and the number of species detected is analogous to the minimal area concept (*Adler et al., 2005*). With increasing trapping efforts, the species richness should level off when the sampling effort (i.e., camera days) is large enough, meaning the inventorying of wildlife species is sufficient. If we know the total species diversity in the area, we then can assess trapping efforts on the species-trapping effort relationship to a certain probability of total species.

In this study, we estimate the MTE on camera traps for species richness following the concept of minimal effort in plant community surveys, using a two-year data set from a small research plot in Gutianshan National Nature Reserve, Zhejiang Province, China. The advantage of using this small, intensely sampled, plot is that we have a high confidence that we detected all the species using the plot during these two years, and thus can subsample our data quantify what types of less intensive sampling would have been efficient. We construct the species-trapping effort relationship by rarefaction of the

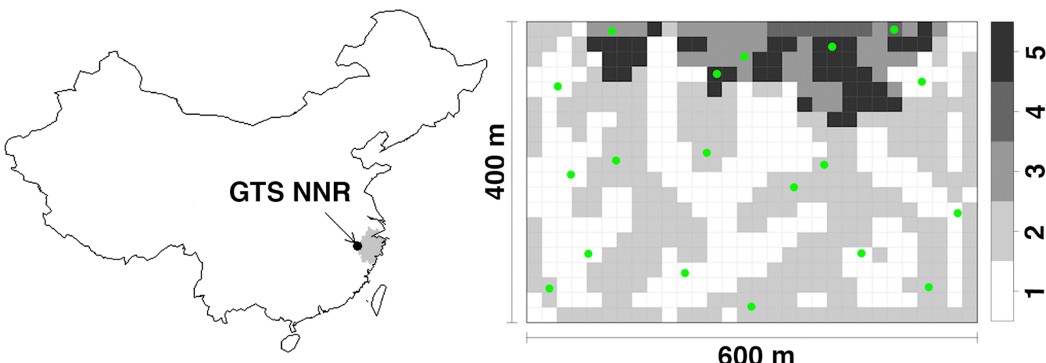

**Figure 1** **The research site.** The location of research plot in Gutianshan National Nature Reserve (GTS NNR) in western Zhejiang Province (grey portion), eastern China, and 19 camera sites (green dots) randomly stratified across habitat types (valleys, ridges, mid-slopes, high slopes and high ridges as numbered from 1 to 5, respectively) in the research plot.

number of terrestrial species added as trapping effort increases, and then evaluate the MTE based on a certain probability of total species. We also evaluate the relative value of adding new camera sites or running cameras for a longer period at one site, and assess the optimal sample period for efficient species detection per camera site.

## MATERIALS & METHODS

### Ethics statement

Our research on camera traps in Gutianshan National Nature Reserve was approved by the Chinese Wildlife Management Authority and conducted under Law of the People's Republic of China on the Protection of Wildlife (August 28, 2004).

### Field site

We deployed camera traps in a permanent plot (400 m × 600 m), which is located in the old-growth forest of the Gutianshan National Nature Reserve (hereafter, the Reserve) (Fig. 1; 29°15′6″–29°15′21″N, 118°7′1″–118°7′24″E) (*Liu et al., 2012*). The protected area of the Reserve is approximately 81 km$^2$, with 57% as natural forest. The Reserve was set up to protect a portion of the old-growth evergreen broad-leaved forest. *Castanopsis eyrei* and *Schima superba* are the dominant species in this forest and in the plot. Most of the forest in our plot is now in the middle and late successional stages (*Legendre et al., 2009*). The annual mean temperature in this region is 15.3 °C and annual mean precipitation is 1964 mm according to the data from 1958 to 1986 (*Yu et al., 2001*). The vegetation is dense and thick with a *c.* 12-m high canopy layer, a rather closed, *c.* 5-m high understory and a dense *c.* 1.8-m high shrub layer (*Chen et al., 2009*). The elevation of our study plot ranges from 446 to 715 m and includes five habitat types in terms of topographic variations: valleys, ridges, mid-slopes, high slopes, and high ridges (*Legendre et al., 2009*).

### Camera positioning

We set 19 infrared digital cameras (Scoutguard SG550; Boly Media Communications Co. Ltd., Shenzhen, China) in the plot at random sites stratified across habitat types from

April 20, 2009 to September 14, 2011 (Fig. 1). The plot was divided into 600 20 m × 20 m subplots. *Legendre et al. (2009)* classified these subplots into five habitat types in terms of topographic variations: valleys (237 subplots), ridges (269), mid-slopes (41), high slopes (8) and high ridges (45) as numbered from 1 to 5 in Fig. 1. Based on the number of subplots of each habitat type, we randomly selected seven, seven, two, one, and two camera sites (all 19 sites) for each habitat type, respectively. We chose the specific positions of camera sites to optimize viewing angle from the tree on which they were mounted. We locked cameras to trees at heights of 40–50 cm above the ground. Cameras faced north or south to reduce the influence of false trigger by the sunrise and sunset. We removed branches or grasses immediately in front of cameras to avoid vegetation from triggering the cameras. A waterproof cover above the camera reduced the effects of rain on our electronics. We did not target animal trails where animal activity is concentrated, and did not use bait or lure to ensure that we only observed the natural movements of animals (*Long et al., 2008*; *Rowcliffe et al., 2011*). We set cameras to take one photograph after each trigger with the interval time of one second. We set all cameras to work 24 h a day and checked memory cards and batteries every month.

## Statistical analysis

Our analyses focus on 19 camera sites with two years of continuous data from June 1, 2009 until May 31, 2011 (730 total days). We excluded seven arboreal birds and squirrels from the analyses (*Tobler et al., 2008*). We also exclude two pheasants (detected one and three times, respectively) and one mammal (detected seven times) in two years, considering these to be nonresident animals passing through the plot from their specialized habitats known to exist in other parts of the Reserve (*Dong, 1990*; *Zhuge, 1990*). We conducted all analyses in R (*R Development Core Team, 2010*).

Individuals of some species will trigger the camera multiple times in a row as they slowly forage at a camera site, resulting in dozens of photographs of the same individual (*Kauffman et al., 2007*). To exclude this influence of replicated photographs triggered by one individual, we set the interval time of 30 min to segregate independent detections of the same species (*Otis et al., 1978*; *O'Brien, Kinnaird & Wibisono, 2003*; *Li et al., 2010*). We used the R function `specaccum` in `vegan` library to conduct our rarefaction analysis (*Oksanen et al., 2013*). The sampling unit in our analyses is one monitoring day. Because we had 19 cameras working simultaneously in the plot, one monitoring day represents 19 camera days (one camera monitoring one day). We then used the rarefaction method to produce the species-trapping effort relationship.

Rarefaction analysis calculates the expected number of species in a small sample of individuals drawn at random from a census or collection (*Simberloff, 1978*; *James & Rathbun, 1981*), and allows for meaningful standardization and comparison of datasets with difference sampling efforts (*Colwell, 2013*; *Gotelli & Colwell, 2001*). So, the rarefaction curve represents a relationship between the number of species (*S*) and the camera days (*D*):

$$S = f(D). \tag{1}$$

We defined the value of $p_s$ as the proportion of common resident terrestrial species (defined common resident species as P2 in Table 1 >1%) in a collection such that

$$p_s = S/S_{max} \qquad (2)$$

where $S_{max}$ is the total number of species. We used the R function `specpool` in `vegan` library (*Oksanen et al., 2013*) through the Chao's method (*Chao, 1987*) to estimate the number of unseen species along with the observed species richness (*Colwell & Coddington, 1994*; *Gotelli & Graves, 1996*). $S_{max}$ was calculated by extrapolating species richness in a species pool:

$$S_{max} = S_{obs} + a_1^2/(2a_2) \qquad (3)$$

which $S_{obs}$ the observed species richness, $a_1$ and $a_2$ are the number of species occurring only in one or only in two sites in the collection. Then we used Eq. (1) to replace $S$ in Eq. (2), and rearranged the formation of Eq. (2) as the proportion of species detected to the number of species richness:

$$f(D) = p_s \times S_{max}. \qquad (4)$$

With the criteria of $p_s$, we carried out trapping efforts from the rarefaction curve.

We conducted a separate analysis to compare the relative value of adding sample days or new sample sites. For each trapping effort (camera days), for example, 10 camera sites monitoring 50 days separately, we sampled 10 camera sites of 19 and continuously sampled 50 days of 730. We calculated the proportion of species in the sampled data. We resampled 1000 times for each trapping effort and obtained the mean value of proportion of species to construct a contour map.

To evaluate how long a camera should be run at one site before cameras should be rotated, we estimated the average values of species richness for each camera site using rarefaction analyses against increasing of monitoring days and independent photographs, respectively.

## RESULTS

### Basic field results

From June 1, 2009 to May 31, 2011, we obtained 3954 photographs of 20 species of animals, 3306 of which could be identified to species, from 1786 independent (>30 min) events during 13,824 camera days (overall photograph rate 0.13 independent photographs/camera day). We excluded seven arboreal species (detected 48 times) and three species (11 times) whose habitats exist in other parts of the Reserve, leaving us with 1727 independent detection events of 10 resident terrestrial species in the plot, including two birds and eight mammals (Table 1). The most photographed species (relative abundance ≥10%) were Confucian niviventer (*Niviventer confucianus*), black muntjac (*Muntiacus crinifrons*), and Reeves' muntjac (*Muntiacus reevesi*). In total these accounted for 74.8% of all independent photographs. The seven arboreal species we excluded were

Peer J

**Table 1 Summary of detected species during two years of camera trap monitoring in Gutianshan research plot.**

| Order/Family | English name | Latin name | RP[*] | IP[*] | P1[*] | P2[*] |
|---|---|---|---|---|---|---|
| **GALLIFORMES** | | | | | | |
| Phasianidae | Chinese bamboo partridge | *Bambusicola thoracicus* | 3 | 3 | 0.17 | |
| | Silver pheasant | *Lophura nycthemera* | 290 | 158 | 8.85 | 9.15 |
| | Elliot's pheasant | *Syrmaticus ellioti* | 48 | 37 | 2.07 | 2.14 |
| | Koklass pheasant | *Pucrasia macrolopha* | 1 | 1 | 0.06 | |
| **PICIFORMES** | | | | | | |
| Picidae | Grey-headed woodpecker | *Picus canus* | 8 | 5 | 0.28 | |
| **PASSERIFORMES** | | | | | | |
| Pittidae | Fairy pitta | *Pitta nympha* | 1 | 1 | 0.06 | |
| Turdidae | Golden mountain thrush | *Zoothera dauma* | 16 | 15 | 0.84 | |
| | White-browed thrush | *Turdus obscurus* | 1 | 1 | 0.06 | |
| | Pale thrush | *Turdus pallidus* | 4 | 4 | 0.22 | |
| Timaliidae | Great necklaced laughingthrush | *Garrulax pectoralis* | 3 | 3 | 0.17 | |
| **RODENTIA** | | | | | | |
| Muridae | Confucian niviventer | *Niviventer confucianus* | 1099 | 826 | 46.25 | 47.83 |
| | Edwards's long-tailed giant rat | *Leopoldamys edwardsi* | 59 | 30 | 1.68 | 1.74 |
| Sciuridae | Pallas's squirrel | *Callosciurus erythraeus* | 30 | 19 | 1.06 | |
| **CARNIVORA** | | | | | | |
| Viverridae | Masked palm civet | *Paguma larvata* | 95 | 89 | 4.98 | 5.15 |
| Mustelidae | Hog badger | *Arctonyx collaris* | 81 | 36 | 2.02 | 2.08 |
| | Small-toothed ferret-badger | *Melogale moschata* | 8 | 7 | 0.39 | 0.41 |
| | Eurasian badger | *Meles meles* | 11 | 7 | 0.39 | |
| **ARTIODACTYLA** | | | | | | |
| Suidae | Wild boar | *Sus scrofa* | 196 | 34 | 1.9 | 1.97 |
| Cervidae | Reeves muntjac | *Muntiacus reevesi* | 643 | 242 | 13.55 | 14.01 |
| | Black muntjac | *Muntiacus crinifrons* | 709 | 268 | 15.01 | 15.52 |

**Notes.**

[*] RP, recorded photographs (*n*); IP, independent photographs (*n*); P1, proportion of independent photographs (%), and P2, proportion of independent photographs without the arboreal and transient animals (%).

grey-headed woodpecker (*Picus canus*: five photographs), fairy pitta (*Pitta nympha*: one), golden mountain thrush (*Zoothera dauma*: 15), white-browed thrush (*Turdus obscurus*: one), pale thrush (*Turdus pallidus*: four), great necklaced laughingthrush (*Garrulax pectoralis*: three), and Pallas's squirrel (*Callosciurus erythraeus*: 19). Three species whose habitats are beyond the plot were also excluded: Chinese bamboo partridge (*Bambusicola thoracicus*: three), koklass pheasant (*Pucrasia macrolopha*: one), and Eurasian badger (*Meles meles*: seven all in one month).

## Minimum trapping effort

In our plot, there were nine common resident species in 10 (Table 1). Therefore, we set $p_s$ as 9/10 = 0.90 to ensure detecting resident species sufficiently. The total number of species estimated by Chao's method was 10. Based on the rarefaction curve (Fig. 2), we replaced the $p_s$ as 0.90, and calculated that we needed 49 monitoring days with our array

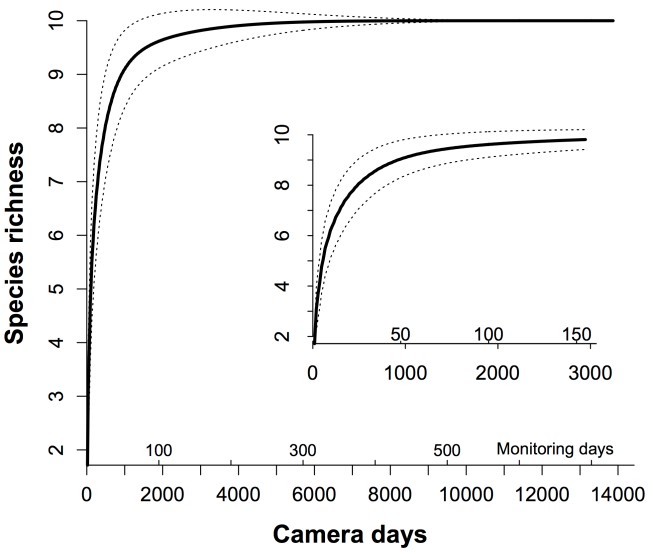

**Figure 2  The species-trapping effort relationship for the communities of the terrestrial animals.** We used rarefaction analysis to generate the species-trapping effort relationship. The 10 resident terrestrial species were detected during 13,824 camera days (see Table 1). Because 19 cameras in our study were working simultaneously in the field, each monitoring day represents 19 camera days. The dashed lines indicate the 95% estimated confidence intervals. The inner graph is an enlargement of the first 150 monitoring days, showing how this relationship increased.

(931 camera days) of the MTE to trap all common resident species efficiently, and $c$. 8700 camera days to trap all 10 residents (Fig. 2).

The proportion of detected species increased rapidly when the trapping efforts were <1000 camera days (Fig. 3). To detect the resident species sufficiently with fewer camera sites, the trapping effort required increases sharply such that more than 2000 camera days would be needed if fewer than three camera sites were used. The contour map of trapping effort shows the pattern of improved detection with more camera sites. Given the same total camera days, it was better to deploy cameras across more sites for a shorter time at each site, than to leave cameras at the same site (Fig. 3). For example, at 1000 camera days (red dashed line), one could have three camera sites at $c$. 350 monitoring days to detect 80% of species, whereas 19 camera sites could detect 90% of species at $c$. 80 monitoring days.

Figure 4 can be used to evaluate how long a camera should be run at one site, showing new species are rapidly detected in the first 40 days (Fig. 4A), or the first 20 independent photographs (Fig. 4B), and then declined.

## DISCUSSION

We take advantage of an exhaustive two-year camera trap survey of one small plot to evaluate biodiversity sampling strategies. We found that 931 camera days of survey would detect 90% of the resident animal species, and that $c$. 8700 camera days would be needed to detect all residents. We were also able to evaluate the question of how long to leave a camera at one site, which is a tradeoff between increasing the probability to detect a species in a site

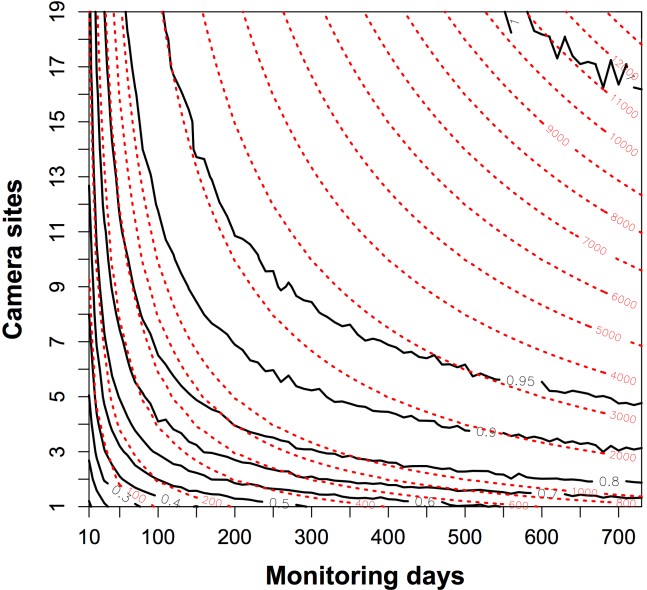

**Figure 3 The contour map of the trapping effort between camera sites and monitoring days.** This map evaluated the relative value of adding more camera sites or more monitoring days in a survey of species diversity. The black bold lines are the proportion of the total species pool ($n = 10$) detected. The red dashed lines show the contour lines of the trapping effort (camera days). The proportion of species detected is mean values resampled 1000 times from a dataset of 19 camera sites running for two years. It showed, given the same trapping effort, it was better to deploy cameras across more sites for a shorter time at each site, than to leave cameras at the same site. See details in the text.

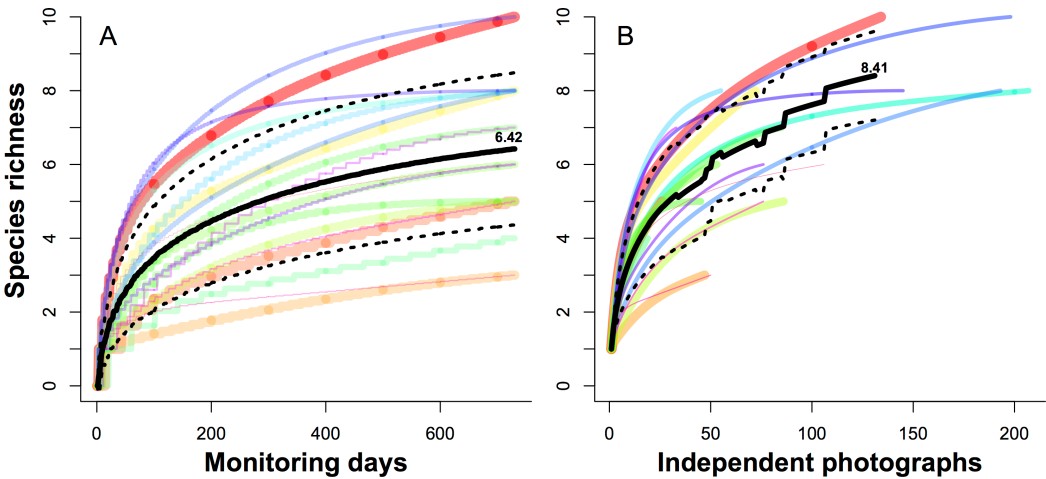

**Figure 4 Species rarefaction curves for 19 camera sites monitored over two years.** Each curve represented the cumulative number of species for each camera site against increasing the monitoring days (A) and independent photographs (B), respectively. The black lines indicate the average values of all 19 camera sites, and dashed lines indicate the 95% confidence intervals.

(the longer, the better) and sampling more sites (the shorter, the better) (*Kays et al., 2009*). Our results showed the density of contour lines along the axis of camera sites were much closer than the axis of monitoring days, meaning deploying more cameras across more sites is a better strategy for species detection. Finally, species rarefaction curves for individual sample sites suggest that *c*. 40 days, or 20 independent photographs, is the optimal sample period for our site, after which the rate of detection of new species declines steeply.

## Rare species

Ten of 20 species in our study were rarely detected, with relative proportions (P1) less than 1%, some even with only one photo (Table 1). Most (35%) of these were common arboreal animals that only occasionally came down to the ground in front of our cameras, and thus were not target species for our study. Two of the very rare pheasants we detected were probably passing through our plot as dispersing or exploring animals. The koklass pheasant we photographed once is known primarily from higher-elevation habitat in the park, while the Chinese bamboo partridge we photographed three times in the lower elevation of the plot was probably visiting from its preferred habitat in nearby farmland (*Zhuge, 1990*). The two rarest terrestrial mammals that we detected were the small-toothed ferret-badger (eight photographs) and Eurasian badger (11 photographs). The records of the small-toothed ferret-badger came regularly throughout the study period, suggesting it is a low-density species of the plot. *Zhang et al. (2009)* had found the hog badger and the small-toothed ferret-badger can co-exist with niche separation. All the records of the Eurasian badger were in March 2010, suggesting it might have been a dispersing animal that visited the plot briefly, so we considered as a non-resident (*Dong, 1990*). All of four rare terrestrial animals we recorded are difficult to observe due to their nocturnal or cryptic activities. Local staff patrol regularly in the reserve, but they rarely record koklass pheasant, small-toothed ferret-badger or Eurasian badger. This shows that camera traps are useful to inventory elusive and rare animals.

## Sampling effort

Other studies using camera trap to monitor wildlife had various MTEs. *Seki (2010)* found 36.3 camera days were needed to survey 90% of forest birds setting the cameras at artificial bathing sites in Japan, but had a very high detection rate of 10.6 independent visits/day. Based on the simulation results, *Rowcliffe et al. (2008)* found usually less than 1000 camera days were needed with at least 20 camera sites that obtained at least 10 photographs per site for the medium-sized mammal in south England. In Azlan's study (*2006*) in a secondary forest in Malaysia, the species accumulative curve leveled off after 16 month with 24 camera stations (*c*. 11,520 camera days) when detecting 25 species of wild mammals (0.36 independent visits/camera day). Due to the different species of interest, the suggested trapping efforts in those researches changed significantly, especially for the animals of large territory and low density. Furthermore, the capture frequencies can also vary between years and sites, so we need a larger trapping effort for a valid survey (*Dorazio et al., 2006*). In our study, there may be several factors resulting in our lower photographic rates (0.13 independent visits/camera day) and higher estimation of trapping efforts (931 camera

days) than other studies. First, the area of our research plot is relatively small (24 ha). Although our plot is a typical habitat in the core-area of the Reserve, it cannot contain all habitat types; some species only use several specific habitats that are beyond our survey area (*Li et al., 2010*). Second, we used fewer (19) camera sites than other studies, which decreases the probability of capturing more animal species in a short time. Finally, we did not put any bait or lures in front of the camera sites and set the cameras in their trails that may also increase the detection rates. As such, we recommended that conservatively 931 camera days were required at the beginning of the experiment design, and suggest future studies in this site rotate camera sites once *c.* 40 days, or after about 20 independent photographs.

## Management implications

Camera traps play an important role in monitoring of biodiversity, and understanding the sampling effort required to reach monitoring goals is important for proper study design (*Soberón & Llorente, 1993*). Although shorter monitoring periods are cheaper and easier, they will also have lower probability of detecting all the species present in an area. Our results from a 19-camera survey of a small plot show 931 camera days, the minimum effort needed to detect the resident terrestrial species at our site. Furthermore, we show that moving cameras more frequently gives more efficient species detections, and show that cameras should not be left at one site for more than *c.* 40 days (or 20 detections).

For long-term monitoring projects, we suggest managers use this approach to find their own MTE from pilot data, while short-term projects should refer to the MTE from projects with similar habitats for guidance. Studies should also consider seasonal activity patterns. Therefore, for short-term projects aiming to inventory the species pool, we should run the projects matching their periods that animals have a high activity, so that we might record the animals easier on cameras in that season.

## ACKNOWLEDGEMENTS

We thank Gutianshan National Nature Reserve for support with fieldwork. We are most grateful to Stuart Pimm for the editorial comments and The Pimm Group at Duke University for their support. We thank Yanping Wang, Guang Hu, Zhifeng Ding, Grant Harris and an anonymous reviewer who provided constructive suggestions to improve the manuscript. We are grateful to Vincent Belluz for language editing on the very early version of the manuscript, Meng Zhang, Qiang Wu for the assistance with fieldwork, and Shusheng Zhang, Yixin Bao for identifying mammal species.

### Funding

The China National Program for R&D Infrastructure and Facility Development (2008BAC39B02), China Scholarship Council (201206320021) and the Fundamental Research Funds for the Central Universities funded this study. The funders had no role

in study design, data collection and analysis, decision to publish, or preparation of the manuscript.

## Grant Disclosures

The following grant information was disclosed by the authors:

China National Program for R&D Infrastructure and Facility Development: 2008BAC39B02.

China Scholarship Council: 201206320021.

Fundamental Research Funds for the Central Universities.

## Competing Interests

The authors declare that there are no competing interests.

## Author Contributions

- Xingfeng Si conceived and designed the experiments, performed the experiments, analyzed the data, wrote the paper, prepared figures and/or tables, reviewed drafts of the paper.
- Roland Kays wrote the paper, reviewed drafts of the paper.
- Ping Ding conceived and designed the experiments, contributed reagents/materials/analysis tools, wrote the paper, reviewed drafts of the paper.

## Animal Ethics

The following information was supplied relating to ethical approvals (i.e., approving body and any reference numbers):

Our research on camera traps in Gutianshan National Nature Reserve was approved by the Chinese Wildlife Management Authority and conducted under Law of the People's Republic of China on the Protection of Wildlife (August 28, 2004).

## Field Study Permissions

The following information was supplied relating to ethical approvals (i.e., approving body and any reference numbers):

The Administration of Gutianshan National Nature Reserve approved our study on camera traps in the research plot in Gutianshan National Nature Reserve (ID GTS2006005).

## Supplemental Information

Supplemental information for this article can be found online at http://dx.doi.org/10.7717/peerj.374.

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
