# Peer review of "How long is enough to detect terrestrial animals? Estimating the minimum trapping effort on camera traps"

_PeerJ, doi:10.7717/peerj.374_

## Round 0.1 · original submission · Minor Revisions

· Academic Editor

Minor Revisions

I think the comments from the reviewers are sensible and that you should be able to make the necessary changes easily. I will need to see in the rebuttal letter all the reviewers' comments and how you have addressed each one of them. If you do this clearly, I can probably make a final decision without sending it back to the reviewers.

Reviewer 1 ·

Basic reporting

The English here is good but not great. Small errors and (mostly) cosmetic changes are necessary to improve clarity and readability. Structure and figures meet standards.

Experimental design

The MS would benefit by discussing in greater detail the uniqueness and advantages of the method it uses (rarefaction….) as opposed to other papers that produce different MTE estimates.

Validity of the findings

No comment

Additional comments

See comments on edited pdf. But in general, I am skeptical of the utility of the results. Camera trap studies have different goals, subjects, and are conducted in various environments, seasons etc. Why should I trust or use these results? Why would these be more broadly applicable than MTE estimates from other studies? Are MTEs even viable for camera trap studies?

Annotated reviews are not available for download in order to protect the identity of reviewers who chose to remain anonymous.

·

Basic reporting

Reporting is fine, but could use some clarification and clarity. Please see detailed comments.

Experimental design

Overall it is fine. Please see comments for suggestions and critique.

Validity of the findings

Fine. The ms would benefit from more explanation. Please see detailed comments.

Additional comments

I appreciate the opportunity to review the ms “How long is enough to detect terrestrial animals? Estimating the minimum trapping effort on camera traps”. Please take all comments constructively, and I hope they serve to improve the manuscript’s message and impact of results.

Overall comments:

1) The Abstract should be updated to explicitly identify what the study did and was about.
2) The study occurs in a large reserve, but sampling is limited to a small plot (24 ha). Therefore, the scale of inference is not for the reserve, but for the plot. This limits the amount of extrapolation attainable from the study.

Review:

Abstract:
1. Abstract states that 20 birds and mammals documented. True, but only 10 used in analyses. Please make this explicit.
2. “to estimate the minimum trapping effort actually needed to sample the wildlife community…in the plot”. I think “in the plot” should be added, for the community in the Reserve is not fully represented. Please make this explicit.

3. The term “resident” should be removed. Evaluating residency, or transiency, did not happen. Instead, state that selection on what species to include in the analyses was based on the # pictures taken, with species having # pictures < # excluded from the analyses.

4. The sentence “At our site, the optimal sampling interval for one site was ca. 40 days or long enough to record about 20 independent photographs.” What does “interval for one site” mean? Perhaps one camera?

This result stems from figure 4 and lines 191-194. But is this really “optimal”. Please make sure this is well explained in the ms (and Abstract). Presently it is not. Sounds as if one could leave 1 camera for 40 days and that would be fine. This is not true for 1 camera, or for all 19 (19 cams is 28 days as depicted in Figure 3 & in text).

5. The sentence “ Analysis shows that rotating cameras to new sites would be a better sampling strategy for measuring species richness than leaving cameras at fewer sites for longer periods” This sentence is not quite true. As I understand it, all cameras were stationary – no rotation happened. What was evaluated is an increasing number of additional stationary cameras were evaluated to determine it’s effect on species richness. This should be reworded in the Abstract to reflect the actual method. Line 68 explained this well.

Introduction – Discussion:

1) Line 83: That habitats in plot are typical of forests should have a reference. Maybe the Legendre et al. 2009.

Now, the assumption is that the species in the plot represent the animal community in the reserve. I don’t think this assumption can be made. As above, I think the species accumulation is for the plot only, not the entire reserve. Please see comment #3 below.

2) Line 94-96: Cameras stratified by habitat types – Figure 1 makes it look like the plots are simply the 18 regions of equal area. Could a sentence (or 2) be added that further explain / summarize this stratification without having to read Legendre et al. 2009? As I understand it, the camera was placed at a random geographical location in each of the 18 sub units. However, Line 124 has 19 cameras and not 18? 18 seemed intuitive based on figure 1, but 19 does not. Please clarify. The ms could use some more explanation on the above.

Optimization was for viewing angle. Makes sense. But cameras were not pointed on trails. Why? If sampling is to maximize capture, why not put them were animals have high activity? What the authors did is not wrong, but wondering why they decided not to acquire pictures, seemingly faster, when they focus on the more common species anyhow.

3) The study occurs in a large reserve, but sampling is limited to a small plot (24 ha). Therefore, the scale of inference is not for the reserve, but for the plot. This should be made explicit. Unfortunately, I think it also limits the amount of inference attainable from the study. Certainly, had more plots been selected, then the authors could have expanded inference to the park. Therefore I think it would help give this paper more punch if the authors described (few sentences) why this plot was chosen out of the entire reserve. Simply to help the results have greater relevance to camera trapping in similar situations &/or this reserve. If I am wrong here, please describe in the ms why I am.

4) It would help to make very clear what the study is attempting to do. For example, MTE on camera traps for species richness. That is explicit. But for what species is not. Line 112 on describes some excluded species, but the study has yet to describe what species are included. That does not come out until the Methods and should be sooner & in the Introduction. Maybe state “common species” defined as “[how you define it would be written in here]”.

5) Now on Line 114 – why exclude the 3 transient animals with occasional visits? Instead, why not include all the species found in the rarefaction?

Line 161 – 164. I still don’t understand why 7 arboreal and 3 “transient” species were removed. Just because few pictures were taken does not mean the species is not a resident. Clearly it is present! Given an objective of this study to provide guidance for camera trapping, how would one know what species are transient or resident in a given study site elsewhere? Certainly, scientists can limit their sample to whatever they choose; I think that by doing so and focusing on more common species limits the study’s utility. For as we know, rare and elusive species usually have fewest images taken and capturing them are often central to such studies. If I am wrong, please explain why.

Perhaps instead of the terms “resident” and “transient” just leave it as “We only considered species that had images taken >=10% of the study in our analyses, because….[we wanted curves to represent those most abundant….or something like this]. And take out the resident / transient verbiage which is subjective.

6) Regarding the independence of images. As I understand it, for species accumulation image independence does not matter, but for rarefaction it does, due to the calculations. I think the ms would benefit by a sentence explaining this. I think this will from an additional way to help educate readers (and if I got this wrong, then me too). And If I got this right, it may make sense to reorganize the Methods, with doing species accumulation first, followed by independent images and then rarefaction.

7) Line 147 – 155: This section of methods somewhat confused me. By looking at the contour map, it seems that 1 camera was sampled for 10 – 700 days (and done repeatedly with the same and other cameras to get mean values [1,000 times]), and then iteratively up to 19 cameras per day. Now the legend reads “because 19 cameras running at once, then each day is 19 camera days”. If that is so, then why does the Y axis have # camera sites listed? And doesn’t 1 camera running for 400 days get nearly 60% of the 10 species? Maybe I am making this harder than it is, but perhaps rewording the methods to make this clearer would help.

It is somewhat implicit that this comes from the rarefaction but I recommend making more explicit.

8) Line 190 – Referring to figure 3: At first I didn’t understand why it is better to have more sites for shorter time than fewer sites for longer time, if the same result is obtained. When I closely examined Fig 3, I got it. For example, at 2000 camera trap days (gray line) one could have 3 cameras at 700 days & 90% of species acquired vs 19 cams at ~100 days and 95% acquired. Quite a big deal! I would state an example or something like this to make Figure 3 more explicit. This is a key figure that you made. It deserves more explanation.

9) Fig 4 needs more explanation too. Looks like each of the cameras is plotted (raw data – accumulation) and then an average accumulation? In keeping with your analyses, why not run the rarefaction for each of the 19 cameras and perform a mean on that?. So, first why look at accumulation and not rarefaction? Isn’t the rarefaction a better predictor and reason for using it? Second, please explain the relevance of why independent photographs matter for predicting species richness in this context and the value of plotting this (panel B). Overall, more text in the ms should clarify the message of Figure 4, and the legend for Figure 4 to explain the significance of 6.42 and 8.6. Looks to me that in “A” if I run one camera in this plot then after 600+ days I will have 6.42 of the 10 species acquired, and that corresponds to the contours on Figure 3…So why even have Figure 4 A?

10) Doesn't specaccum provide standard deviation? I don't necessarily think it relevant unless Figure 4 retains accumulation and not rarefaction. If it retains accumulation, and a multiple runs ensued then a sense of variability is important and should be included. Admittedly, this comment stems from some confusion about Figure 4, what it represents, and what it is trying to accomplish.

10) Paper could use minor grammatical checking throughout.

---

## Round 0.2 · accepted · Accept

· Academic Editor

Accept

I have no additional comments to make: I appreciate your taking care of the reviewers' comments.